

# Observation of the rock slope thermal regime, coupled with crack meter stability monitoring

Ondřej Racek [1,2], Jan Blahůt [1], Filip Hartvich[1]

[1] Institute of Rock Structure and Mechanics, Czech Academy of Sciences, Department of Engineering Geology, V
Holesovickach 94/41, 182 09, Prague, Czechia
[2] Charles University in Prague, Faculty of Sciences, Department of Physical Geography and Geoecology, Albertov 6, 128 43,
Prague, Czechia

*Correspondence to*: Ondřej Racek (racek@irsm.cas.cz)

**Abstract.** This article describes an innovative, complex and affordable monitoring system designed for joint observation of
environmental parameters, rock block dilatations and temperature distribution inside the rock mass with a newly designed 3-
meter borehole temperature sensor. Global radiation balance data are provided by pyranometers. The system introduces a novel
approach for internal rock mass temperature measurement, which is crucial for the assessment of the changes in the stress field
inside the rock slope influencing its stability. The innovative approach uses an almost identical monitoring system at different
sites allowing easy setup, modularity and comparison of results. The components of the monitoring system are cheap, off-the-
shelf and easy to replace. Using this newly designed system, we are currently monitoring three different sites, where the
potential rock fall may endanger society assets below. The first results show differences between instrumented sites, although
data time-series are relatively short. Temperature run inside the rock mass differs for each site significantly. This is very likely
caused by different aspects of the rock slopes and different rock types. By further monitoring and data processing, using
advanced modelling approaches, we expect to explain the differences among the sites, the influence of rock type, aspect and
environmental variables on the long-term slope stability.

**Keywords:** monitoring, rock slope, stability, temperature, crack meter, horizontal borehole temperature

## 1 Introduction

The rock slope stability is crucially influenced by both endogenous rock properties and exogenous factors (D'Amato et al.,
2016, Selby 1980). The rock physical properties are well known and numerous laboratory experiments and theoretical works
exist in the field, however, there are very few in-situ experiments that would deal with real-world time and space scales (Fantini
et al., 2016; Bakun-Mazor et al., 2013, 2020; Janeras et al., 2017; Marmoni et al., 2020)

Thermal expansion and frost action are the main exogenous physical processes of the mechanical weathering of the rock
surface, which together with chemical weathering ultimately results into a weakening of the rocks slopes and lowering their
stability (Gunzburger et al., 2005, Vespremeanu-Stroe and Vasile, 2010; do Amaral Vargas et al., 2013; Draebing, 2020). The
loss of stability, caused by repeated changes in the stress field inside the rock eventually leads to a rockfall, one of the fastest


and most dangerous forms of slope processes (Weber et al., 2017; Gunzburger et al., 2005). In the alpine environment, rock falls are increasingly caused by permafrost degradation and frost cracking (Gruber et al., 2004; Ravanel et al., 2017) or temperature related glacial retreat (Hoezle et al., 2017). To address the influence of permafrost melting on the rock slope stability, several monitoring systems/campaigns were proposed. Magnin et al., (2015a) constructed a monitoring system consisting of rock temperature monitoring both on the rock face and in-depth sensors. In-depth rock mass temperature monitoring is placed in up to 10m deep boreholes. The monitoring is coupled with ERT campaigns to determine sensitive permafrost areas (Magnin et al., 2015b). Girard et al., (2012), introduced a custom acoustic emission monitoring system for quantifying freeze induced damage in rock.

Among the destabilizing processes caused by changes in rock temperature and contributing to the decrease of stability are: rock wedging-ratcheting (Bakun Mazor et al., 2020; Pasten et al., 2015), repeated freeze-thaw cycles, thermal expansion-induced strain (Gunzburger et al., 2005; Matsuoka 2008) and in specific conditions, exfoliation sheets can be destabilized by cyclic thermal stress (Collins and Stock, 2016; Collins et al., 2017). These processes can be repeated many times in specific weather conditions, thus effectively widening the joints and fracturing the rock. Unfortunately, the last two winters in Czechia were relatively warm, which is not ideal for observing the freeze-thaw cycles. To counter this, we are currently preparing new installation in the Krkonoše Mountains (northern Czechia) at the altitude of 1270 m a.s.l. where also rock wedging-ratcheting process should be active because of the suitable disposition of newly instrumented blocks.

Rock slope monitoring is one of the common tasks in engineering geology, often directly connected to the safety of large construction sites, such as dams, power plants, bridges, or tunnels (Ma et al., 2020, Li et al., 2018; Scaoni et al., 2018). Monitoring of rock slope stability can be designed using various approaches, with a background in geodesy, using GNSS or total station (Gunzburger et al., 2005; Reiterer et al., 2010; Yavasoglu et al., 2020), geotechnics with crack meters, inclinometers and extensometers (Greif et al., 2017; Lazar et al., 2018), geophysics with ambient vibration monitoring, ERT profiling and micro seismical sensors (Burjanek et al., 2010; 2018; Weber et al., 2018; Coccia et al., 2016; Yan et al., 2010; Weigand et al., 2020; Warren et al., 2013), or remote sensing methods, such as TLS (Terrestrial Laser Scaner), UAV (Unmanned Aerial Vehicle) or ground based photogrammetry or GB InSAR (Ground Based Interferometric Synthetic Aperture Radar) (Sarro et al., 2018; Matano et al., 2015).These systems using various types of sensors (Fantini et al., 2017, Janeras et al., 2017). Measurement of air temperature, accelerometers, cameras and seismographs are often used to detect and explain rock fall events (Burjanek et al., 2010, 2018; Tripolitsiotis et al., 2015; Matsuoka, 2019). The use of these methods is more suitable for monitoring larger parts of rock slope and allows spatiotemporal identification of rock fall events. On the other hand, tiltmeters and extensometers are usually used to single unstable block/part of rock slope monitoring (Barton et al., 2000; Lazar et al., 2018). These point measuring methods can describe spatial changes of a monitored feature with higher accuracy, however, the use of these devices does not allow to monitor larger parts of rock slope.

Complex monitoring systems are used to monitor potentially unstable rock slope parts. Janeras et al., (2017) introduced a multi-technique approach of rockfall monitoring of unstable mass. The system consists of crack meters, TLS, GB InSAR and total station surveying. Jaboyedoff et al., (2004) used geodetic network, extensometer network and weather monitoring. Vaziri




et al., (2010) presented a review of monitoring techniques for open-pit mine walls monitoring. Carla et al., (2017) used GB
InSAR to monitor displacement of mine slopes failures. Large rock slides are monitored by Crosta et al., (2017) using GB
InSAR, Satelite InSAR and borehole inclinometry. Loew et al., (2012) used borehole inclinometry and borehole extensometry
combined with GB InSAR interferometry in the large Randa rockslide monitoring. Zangerl et al., (2010) used total station
measurements, coupled with borehole inclinometers for a similar purpose. Long-term rock slope destabilization is monitored

using total station measurements, multipoint surface extensometers, borehole inclinometers (Chen et al., 2017), or TLS
measurements eventually (Hellmy et al., 2019). Usually, these monitoring systems are designed as experimental, aiming to
develop new early-warning sensors or approaches (Loew et al., 2017; Jaboyedoff et al., 2011) or to describe processes of rock
slope destabilization (Fantini et al., 2016; Kromer et al., 2019; Du et al., 2017). However, these systems are site-specific and
installation of a similar system on more sites is complicated and financially demanding.

These systems are sometimes complemented with environmental data observations. However, these are often limited to air
temperature and/or rock face temperature monitoring only (Jaboyedoff et al., 2011, Blikra and Christiansen, 2014; Marmoni
et al., 2020; Collins and Stock, 2016; Collins et al., 2018; Eppes et al., 2016). Less commonly, the temperature is measured in
rock mass depth (Magnin, et al., 2015a, Fiorucci et al., 2018). The absence of precise data about temperature changes in rock
mass depth makes the assessment of the thermally-induced stress field response inside the rock mass complicated. Without in-

depth temperature data and incoming radiation, the determination of heating/cooling trends causing internal volume and stress
field changes is difficult. Also, the monitoring systems are usually designed specifically for the monitored sites, which brings
difficulties for generalization of the results or installation of the system at more sites.

Therefore, we have designed an easy to modify monitoring system, which measures the physical parameters in a 2D
environment in the field conditions, both on the rock face and inside the rock mass. With just minor modifications we can

instrument various rock slope sites.

## 2 Monitoring methods

The monitoring methods mentioned in introduction have recently gone through a massive development concerning precision,
accuracy, reliability, sampling rate, and applicability. Even completely new methods were established, for example, UAV
applications, TLS, etc. This expansion was mostly allowed by the rapid development of corresponding fields of informatics,

computation technologies, communication channels and satellite technology applications.

Unlike to above-mentioned systems, the monitoring system which we are presenting (Fig. 1), can be placed at various sites
without major modifications. Using common safety rules and methods for working in heights, the system can be placed directly
within vertical or even overhanging rock face. Anchoring for system installation must be placed within a stable part of the rock
slope, which ensures worker´s safety under any circumstances. This monitoring design brings an opportunity to compare results

from different locations and observe generally applicable regularities in rock face behaviour thanks to similar monitoring
methodology. Our monitoring system (Table 1, Fig. 1) is composed of the following components:





- a set of automatic induction crack meters, coupled with dataloggers (Fig. 1) measuring relative block displacement

- a environmental station with a set of sensors measuring various meteorological data (Fig. 1), such as air temperature, humidity and pressure (Table 1), and global radiation balance of the rock face (Fig. 4) using pair of pyranometers

- a set of 12 thermometers placed along a 3 m deep borehole, carefully insulated between each neighbouring sensors, measuring rock slope thermal depth profile at ten minutes interval

| Component | Manufacturer | Accuracy | Resolution | Repeatibility | Measuring range | Max sampling rate | Protection | Operational temperature | Service life | Price |
|-----------|-------------|----------|------------|---------------|-----------------|-------------------|------------|------------------------|--------------|-------|
| Crackmeter Gefran PZ 67-200 | GEFRAN (It) | <0.1 % | 0.05 mm | 0.01 mm | 0-200 mm | N/A | IP67 | -30 - 100 °C | > 25*10^8 m strokes | 300 € |
| Datalogger Tertium Beacon | Tertium tech. (It) | N/A | N/A | N/A | N/A | < 1 sec | IP65 | -30 - 60 °C | > 5 years | 190 € |
| Datalogger Temp. Sensor | Tertium tech. (It) | 0.02 °C | 0.01 °C | N/A | -30 - 60 °C | < 1 sec | IP67 | -30 - 60 °C | > 5 years | |
| Control unit, battery, solar panel | FIEDLER (Cz) | N/A | 0.00X; 16bit | N/A | N/A | 1 min | IP66 | -30 - 60 °C | > 5 years | |
| Temperature sensor | FIEDLER (Cz) | 0.1 °C | 0.1 °C | 0.01° C | -50 - 100 °C | 1 min | IP66 | -50 - 100 °C | > 5 years | |
| Rain gauge SR03 500cm2 | FIEDLER (Cz) | 0.05 mm | 0.1 mm/year | 0.1 mm | N/A | 50 m. sec | IP66 | 0 - 60 °C | > 5 years | 1 650 € |
| Humidity sensor | FIEDLER (Cz) | 0.008 % | < 0.1 %/year | 0.02 % | 0 - 100 % | 1 min | IP66 | -50 - 100 °C | > 5 years | |
| Atmospheric pressure sensor | FIEDLER (Cz) | 2 mbar | 0.025 mbar | 0.1 mbar | 300 - 1100 mbar | 1 min | IP66 | -40 - 70 °C | > 5 years | |
| Pyranometer SG002 | Tlusťák (Cz) | 10%/day | 20 µV/Wm² | < 5% | 300 - 2800 nm (0 - 1200 W/m³) | 1 min | IP66 | -30 - 60 °C | > 5 years | 450 € |
| Borehole temperature sensor | FIEDLER (Cz) | 0.1 °C | 0.1 °C | 0.01° C | -50 - 100 °C | 1 min | sealed inside bh | -30 - 60 °C | > 5 years | 1 150 € |
| Datastorage/procesing | FIEDLER/SigFox | / | / | / | / | 1 hour | / | / | infinite | 200 € |

**Table 1: List of presented monitoring system components, with performance metrics and prices.**

All the elements of the system are commercially available at affordable expenses (one site instrumentation costs approx. 5000 Eur (Table 1), and are easy to replace by even moderately experienced user. Additional costs are drilling works (1-2 000 EUR). Cost of drilling works depends on the site accessibility and rock mass hardness. The price of the specific
monitoring system is also affected by the number of used crack meters and data loggers. On the other hand, system maintenance costs are not higher than 300 Eur per year including data processing and storage.





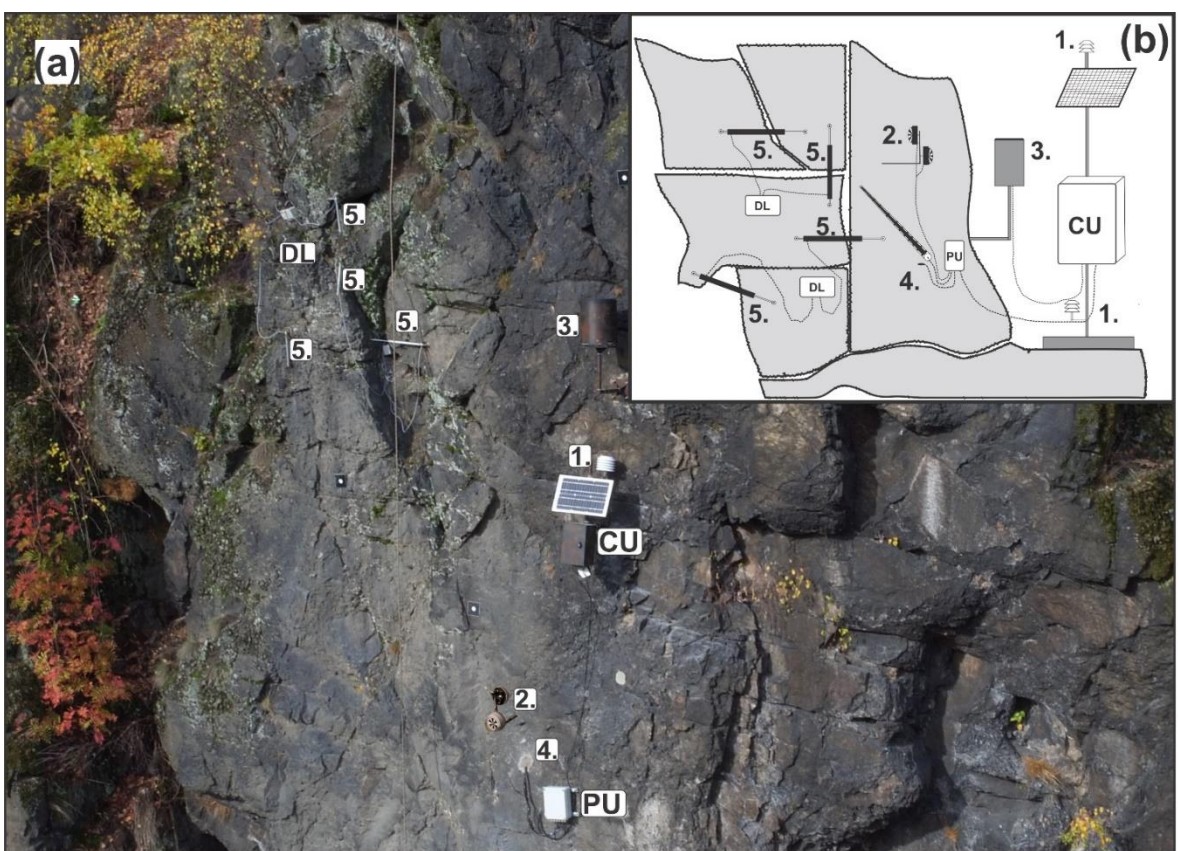

**Figure 1: Photo of actual monitoring system at Tašovice site (a). Generalized scheme of the monitoring system (b). CU: control unit, PU: processing unit, DL: data logger, 1.: Temperature sensor, 2.: Pyranometers, 3.: Rain gauge, 4.: Borehole temperature monitoring, 5.: Crack meters (only four of total six crack meters are visible on this photo)**

## 2.1 Dilatation monitoring

At each site, suitable joints separating unstable rock blocks were selected. Joints were selected to best represent general directions of expected rock blocks movements. Where it was possible, joints that directly separate unstable block from rock slope were chosen. These joints were afterwards instrumented with automatic crack meters Gefran PZ-67-200, working on the induction principle. These crack meters are suitable for harsh conditions, where are affected by temperature changes, snow cover, ice accumulation or rainfall. The protection level of crack meters is IP67. These crack meters work with good measurement accuracy (Table 1) (GEFRAN, 2019). Crack meters are coupled with Tertium TAG dataloggers (Tertium technology, 2019), which also contain accurate in-situ temperature sensors (Table 1). When a datalogger is placed within the discontinuity, the local temperature microclimate can be estimated. The joint dilatation and temperature data are stored in the datalogger and can be wirelessly transmitted at a distance of up to a hundred meters using wi-fi, which simplifies data collection as it can be usually performed from below the rock face. Tertium TAG data can be sent to a server via IoT SigFox network. The crack meters and dataloggers are powered with two AA batteries, which last typically 6-12 months. The displacement and



temperature are set to be measured every hour. This can be however remotely changed if necessary, for example during special experiments such as thermal camera monitoring campaigns (Racek et al., 2021).

## 2.2 Environmental monitoring

For the monitoring of the environmental and climatic parameters at the study sites, we use automatic environmental stations manufactured by Fiedler environmental systems. These are composed of registration, communication and control unit M4016-G, external tipping-bucket rain gauge, two temperature sensors, atmospheric pressure sensor, humidity sensor, and a pair of pyranometers, measuring the global radiation. All these sensors and the control unit are powered by a 12 V battery, which is charged by a small solar panel (Fig. 1). Except for precipitation, which is measured continuously using a pulse signal, all other climatic variables are measured every 10 minutes. The control unit is equipped with a GSM modem, which sends the data automatically to the server of the provider every day. For information about accuracy, durability and price of environmental monitoring see table 1. To expand the spatial extent of temperature data, thermal camera time-lapse campaigns were performed and are also planned in future (Racek et al., 2021).

To compute the radiation balance of a rock face, it is necessary to measure both incoming and reflected radiation. For this purpose, a set of pyranometers is used (Gunzburger and Merrien-Soukatchoff, 2011; Janeras et al., 2017; Vasile and Vespremeanu-Stroe, 2017). Our monitoring system uses two pyranometers placed perpendicular to the rock face, one facing the rock surface while the other the sky hemisphere. This setup enables to measure both incoming and reflected radiation. The sensors are not placed directly on the rock face, but on an L-shaped holder, which allows placing both sensors almost at the same point. The rock-facing pyranometer is placed at a distance of approx. 10 centimetres from the rock surface. The pyranometers (type SG002) are supplied by Fiedler environmental systems company (FIEDLER, 2020), and have an output of 0–2 V, which corresponds to global radiation of 0–1200 W/m$^3$, the monitored wavelength spans from 300 to 1200 nm. Outputs from pyranometers are processed by a converter and then send to the control unit, to be sent with the other monitored meteorological variables to the data hosting server.

## 2.3 Borehole temperature monitoring

For the complex monitoring of the thermal behaviour of a rock slope, it is necessary to know temperatures at different depths of the rock mass. This is a crucial and innovative part of our monitoring system. Temperature from rock mass depth contributes to a better understanding of the rock slope thermal regime.

The sensors are placed in a 3 m deep borehole. The borehole is drilled close to the monitored unstable rock blocks. However, to ensure safety during drilling and the long lifespan of borehole and sensors, the borehole itself is drilled to the stable part of the rock slope, perpendicularly to the surface. The borehole is then equipped with a custom-designed device with a set of temperature sensors, placed along a tubular spine at different depths. Technical parameters of temperature sensors are the same as for air temperature sensors (Tab 1). Copper rings with 5 cm diameter, connected to thermal sensors, are placed at a given distance on the tubular spine (5 cm below the surface, 10 cm, 20 cm, 30 cm, 50 cm, 75 cm, 100 cm, 150 cm, 200 cm,


250 cm and 300 cm). Additionally, one temperature sensor is placed directly on the rock surface. The head of the borehole is insulated, to prevent air and water inflow into the rock, and the sensors inside the borehole are separated by thorough thermal insulation, to ensure the temperatures are not affected by the air circulation in the borehole. The thermal data, collected every 10 minutes, are passed through a converter and send to the main control unit of the environmental station.

## 3 Instrumented sites

The monitoring system has been so far established at three different sites (Fig. 2), using the same instrumentation set-up. The sites were chosen deliberately in steep rock slopes built of various rock types, with various aspect, diverse geological history and, to integrate a practical applicability side, at locations where the potential rockfall endangers buildings, infrastructure or other social assets.

**Figure 2: Overview of three so far instrumented sites. Position of atmospheric monitoring and 3 m thermometer borehole. By different colours are indicated monitored rock blocks.**



### 3.1 Pastýřská rock (PS)

The first instrumented rock slope (Fig. 2) called "Pastýřská rock" is located on the Elbe riverbank in Děčín town, NW Czechia.
Monitoring of meteorological variables was started in late 2018, followed by crack meters installation and in-depth borehole
temperature sensor. Pastýřská rock is formed by Cretaceous sandstone, with a general southeast orientation. The mechanical
and physical properties of sandstone samples are listed in table 2. The rock slab with pyranometers and borehole is dipping
87° towards the east (85°). On this site, using traditional methods, three main discontinuity sets were identified (Table 3). This
locality was known for extensive rock fall activity in past, which lead to rock slope stabilization works in the late 1980s.
However, the block monitored by the crack meters remained in its natural state, without any stabilization measures. At this
site, one block is monitored, using two pairs of crack meters. This partial block has dimensions of 6.7 x 10.7 x 2.5 m.
Monitoring at this rock slope has been in operation since autumn 2018. The monitored block is located in the overhang part of
the rock slope and all four visible cracks are monitored. The colour of the rock slope surface varies from dark, to light grey
(Fig. 2). The rock slab, where the pyranometers are placed is coloured in light grey colour.

### 185 3.2 Branická rock (BS)

This rock slope (Fig. 2) in Prague (Central Czechia) was instrumented in summer 2019 and is formed by several Silurian and
Devonian limestone layers, with varying mechanical and physical properties (Table 2). The rock slope was artificially created
and used till the 1950s as a limestone quarry. The rock slope is located on a Vltava riverbank and it is generally facing west-
south-westwards. The pyranometers and the borehole temperature sensors are placed on a rock slab dipping 80° to the
southwest (235°). Three main discontinuity sets were identified using a geological compass at Branická rock site (Table 3).
The site was known for extensive rock fall activity in the past, even after quarry closing, which resulted in partial stabilization
of most unstable blocks in the 1980s. At this site, three unanchored blocks (Fig. 2) are monitored with seven crack meters. In
the upper part of the rock slope lies the largest monitored block at this site, with dimensions 0.9 x 4.5 x 3.7 m. This block is
monitored with three crack meters. The second block is located at the bottom part of the rock slope, partly shaded by vegetation.
Dimensions of the second block are 2.5 x 1.6 x 3.6 m. The second block slowly slides on the bottom surface and is instrumented
with two crack meters. Finally, the third monitored block is smaller (0.8 x 1.4 x 0.4 m). It is located in a highly weathered part
of rock slope and monitored with two crack meters. Monitoring at Branická rock site is running since autumn 2019. The colour
of limestone varies from grey to yellow (Fig 2) and the colour of limestone facing pyranometer is light grey.

### 3.3 Tašovice (T)

The third instrumented site (Fig. 2) is a rock slope above a local road and Ohře river near Karlovy Vary town, W Czechia.
Rock slope is formed by partly weathered granite with varying mechanical and physical parameters (Table 2). Generally, it is
facing south-south-east direction. The instrumented slab is dipping 88° to the south (170°). At this site, three relatively poorly

developed discontinuity systems were identified using a geological compass in the field (Table 3). At this site, small rock falls are frequent as it can be seen from the fresh rock and debris accumulation under the rock face. The locality was fully

instrumented with borehole temperature sensors, environmental station and global radiation monitoring in spring 2020. Three relatively small blocks are monitored at this site. Block 1 (1.7 x 1 x 2.1 m), Block 2 (0.9 x 0.8 x 0.4 m) and Block 3 (0.5 x 1.2 x 0.4 m). Each block movement is monitored with a pair of crack meters. The colour of the rock slope varies from black to dark grey. The granite surface at the pyranometers site has dark grey colour.

## 4 Fieldwork campaigns

Each instrumented rock slope was characterized using traditional geological, geomorphological and geotechnical methods, such as measuring geometrical properties of joints and fault planes, relative surface strength measurement using a Schmidt hammer, discontinuity density measuring, and stability assessment estimated using geotechnical classifications (Racek, 2020). Mechanical and physical properties of the rocks were determined by common laboratory tests, using collected representative rock samples (Table. 2).

| site | samples | ultrasound testing (wet samples) | | | | | pressuremeter (dry samples) | | | | | Brazilian test (dry samples) | |
|---|---|---|---|---|---|---|---|---|---|---|---|---|---|
| | | ρ [g/cm³] | E [GPa] | v [GPa] | v | K[GPa] | hardness [MPa] | E [GPa] | μ [GPa] | v | K [GPa] | Fmax [kN] | σrt [MPa] |
| Pastýřská rock - sandstone | unweathered | 1.870 - 1.926 | 13.8 - 17.4 | 5.8 - 7.7 | 0.12 - 0.26 | 6.6 - 10.4 | 22.3 - 28.5 | 14.8 - 17.2 | 6.2 - 6.9 | 0.17 - 0.24 | 7.6 - 11.2 | 3.0 - 5.5 | 1.3 - 2.4 |
| | weathered | 1.810 - 1.991 | 8.5 - 15.8 | 3.7 - 6.3 | 0.14 - 0.28 | 4.1 - 11.9 | 3.9 - 11.0 | 2.2 - 6.0 | 1.0 - 2.4 | 0.24 - 0.39 | 3.9 - 4.0 | 0.7 - 3.6 | 0.3 - 1.6 |
| Branická rock - limestone | unweathered | 2.689 - 2.698 | 75.1 - 79.6 | 29.2 - 30.8 | 0.28 - 0.29 | 58 - 61.9 | 77.1 - 244.6 | 65.8 - 75.0 | 24.9 - 29.0 | 0.28 - 0.41 | 50.7 - 129.7 | 14.1 - 36.1 | 5.9 - 15.6 |
| | weathered | 2.678 - 2.698 | 73.4 - 78.1 | 27.9 - 30.2 | 0.29 - 0.34 | 62.2 - 64.3 | 88.2 - 170.5 | 63.6 - 73.1 | 24.4 - 28.2 | 0.27 - 0.31 | 49.3 - 61.0 | 18.1 - 33.4 | 7.8 - 14.0 |
| | with cracks | 2.675 - 2.697 | 64.5 - 78.4 | 24.4 - 30.3 | 0.29 - 0.32 | 60.4 - 63.4 | 52.1 - 192.3 | 25.4 - 74.0 | 9.6 - 27.9 | 0.27 - 0.33 | 24.7 - 61.2 | 11.4 - 26.9 | 4.7 - 10.9 |
| Tašovice - granite | weathered | 2.399 - 2.525 | 5 - 11.9 | 1.8 - 4.2 | 0.39 - 0.42 | 7.6 - 22.7 | 36.1 - 63.1 | 4.3 - 15.0 | 1.6 - 5.6 | 0.27 - 0.41 | 4.4 - 20.4 | 6.5 - 11.2 | 2.4 - 5.0 |

**Table 2: Mechanical and physical properties of laboratory tested rock samples from all three monitored sites.**

Traditional methods are supplemented with state-of-the-art methods of rock slope analysis, including analyses of 3D point clouds and derived mesh surfaces, based on SfM (structure-from-motion, a computerized photogrammetric technique

based on the calculation of 3D point cloud from overlapping photos with varying focal axis orientation) (Westoby et al., 2012) processing using the data collected with a UAV or TLS collected data. The obtained detailed rock surface models are then analysed using Cloudcompare and its plugins (Girardeau-Monaut, 2016; Thiele et al., 2018; Dewez et al., 2016) and DSE software (Riquelme et al., 2014) to derive the joint and fault planes and measure their spatio-structural properties. Moreover, three main discontinuity systems that were identified using a geological compass in the field at all three sites are summarized

in Table 3.

| Discontinuity sets | Pastýřská rock | Branická rock | Tašovice |
|---|---|---|---|
| Set 1 | 80/40 | 50/325 | 50/90 |
| Set 2 | 86/310 | 90/197 | 50/220 |
| Set 3 | 80/275 | 62/85 | 88/345 |
| Set 4 | 180/30 | 62/210 | 46/181 |



**Table 3: Three main discontinuity sets identified in the field using geological compass. Dip/Dip direction.**

Point cloud data, that were produced by UAV SfM photogrammetry were analysed, edited in Cloud Compare and afterwards principal poles (Fig. 3) were automatically identified using DSE software (Riquelme et al., 2014).

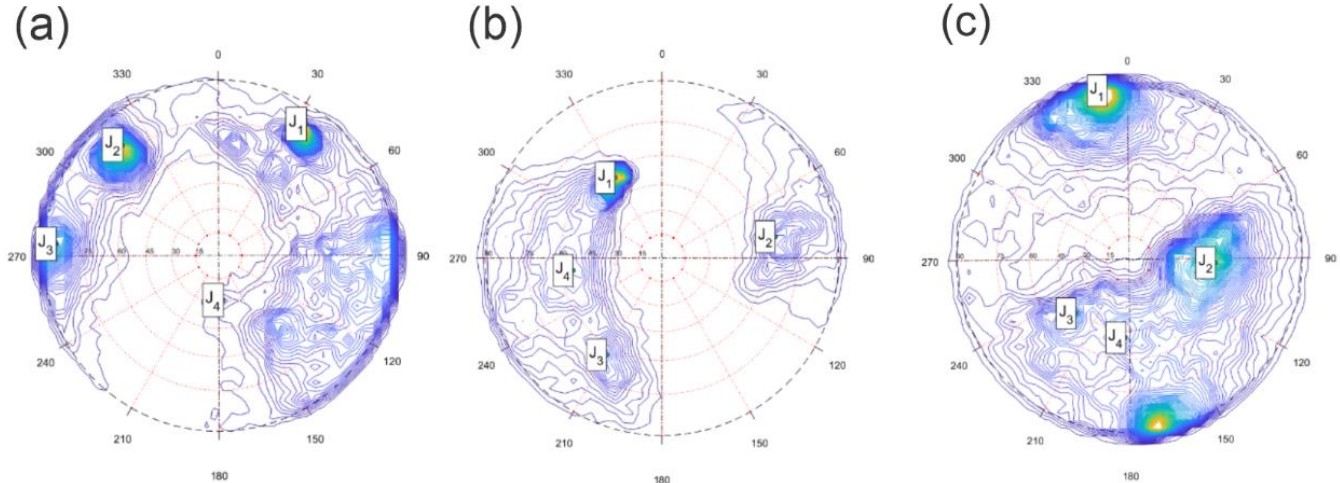


**Figure 3: Principal poles, with four main discontinuity sets (J1 – J4) classified using DSE software (Riquelme et al., 2014). Density of principal poles corresponds to main discontinuity sets identified from pointclouds. (a): Pastýřská rock, (b): Branická rock, (c): Tašovice.**

## 5 First results

The monitoring systems are operational from 1 to 2 years. During most of the period, the gauges and sensors operated without problems or interruptions. However, some accidents or breakdowns occurred, the most serious being the destruction of one pyranometer by boulders, washed down by a rainstorm. As the experimental sites are easy to reach and spare parts easy to obtain, any broken or damaged elements can be replaced within a few days. Workers within rock faces are using safety gear, such as full-body harness and helmet. Securing is done with static ropes and working grade brake. In the case of Pastýřská

rock, workers can use Via Ferrata routes.

### 5.1 Environmental monitoring

Environmental monitoring on all instrumented sites works without problems. From measured time-series of meteorological variables (Table 4) rock slope microclimate can be defined. Also, the influence of these on monitored discontinuities position can be determined. Comparison of crack opening with measured rainfall events using simple graph does not indicate any

visible influence of precipitation on the crack opening/closing. However, the measuring period is still short, with prevailing dry, relatively warm weather. Conversely, there is a visible influence of air temperature to block dilatation, where both diurnal and annual cycles can be identified (Fig. 6). Basic statistical data descriptions of measured environmental variables are listed in Table 9.





| Site | Active since | Active days | Rainfall [mm] sum | Temp. [°C] min | max | mean | Pressure [hPa] min | max | mean | Humidity [%] min | max | mean |
|------|--------------|-------------|-------------------|----------------|-----|------|--------------------|-----|------|------------------|-----|------|
| Pastýřská rock | 25.01.18 | 1098 | 1503 | -9 | 41.2 | 10.5 | 963.5 | 1026.4 | 997 | 13.3 | 96.1 | 72.9 |
| Branická rock | 21.05.19 | 617 | 1020 | -7 | 44 | 13.2 | 955.3 | 1017.4 | 987 | 12.5 | 95.8 | 70.4 |
| Tašovice | 12.12.18 | 777 | 691 | -10 | 45.5 | 10.4 | 935.3 | 997.1 | 969 | 17.4 | 96.7 | 76.4 |

**Table 4: Overview of measured meteorological variables at all three sites. The last measurements considered were measured 27.1.2021.**

### 5.2 Rock surface radiation balance

Monitoring of rock face radiation balance was installed at monitored rock slopes during 2020, therefore we still miss a full-year global radiation cycle. Even from these incomplete data we can observe the differences between individual sites (Fig. 4).

Basic statistical description of so far measured data is listed in table 5. Local conditions influence incoming radiation pattern by general aspect of the rock slope or by shading effect of pyranometer´s surroundings. Differences in the absolute reflected radiation are mainly caused by the different colour of rock faces, by different heating and cooling trends of the rock mass and by the different angle of incoming solar radiation caused by the aspect of the instrumented slab.

| | Radiation incoming [W/m$^2$] | | | Radiation reflected [W/m$^2$] | | |
|-----------|----------------|---------------|----------|----------------|---------------|----------|
| | Pastýřská rock | Branická rock | Tašovice | Pastýřská rock | Branická rock | Tašovice |
| **Mean** | 85.9 | 156.7 | 34.9 | 25.0 | 85.9 | 2.9 |
| **Variance** | 38896.8 | 53102.1 | 8910.6 | 2304.4 | 10907.8 | 111.5 |
| **Stdev** | 197.2 | 230.4 | 94.4 | 48.0 | 104.4 | 10.6 |
| **Median** | 0.0 | 20.3 | 9.4 | 0.0 | 0.0 | 0.0 |
| **$Q_3$** | 54.5 | 189.6 | 20.3 | 15.0 | 160.0 | 1.1 |
| **$Q_1$** | 0.0 | 4.6 | 8.7 | 0.0 | 2.3 | 0.0 |
| **Min** | 0.0 | 0.0 | 0.0 | 0.0 | 0.0 | 0.0 |
| **Max** | 1198.8 | 1197.3 | 1052.3 | 413.9 | 910.0 | 119.1 |

**Table 5: Basic statistical description of pyranometers measured data.**



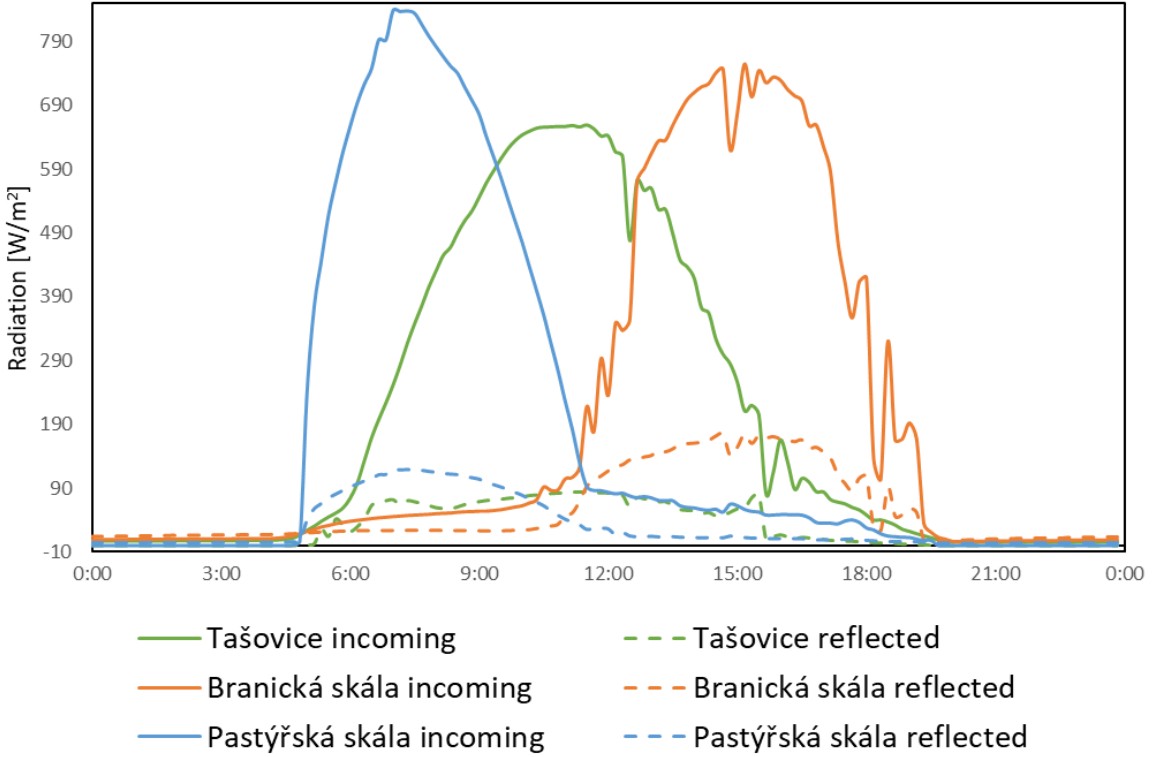

**Figure 4: Example of the incoming and reflected radiation measured by pyranometers at BS, T and PS sites. 24-hour time series of incoming and reflected radiation. Data were recorded 1.8.2020 with no clouds. Influence of slope aspect is obvious from peak incoming radiation shift.**

### 5.3 Borehole temperature

By continuous temperature measuring in different depths inside a horizontal borehole, we can observe both diurnal and annual temperature amplitude in various depths (Fig. 5). In-depth measurements of temperature show differences in temporal thermal behaviour between monitored rock slopes (Fig. 5). From boxplots that represents data from all monitored sites (Fig X.), it is obvious that largest surface temperature variation has been measured at Tašovice site despite the shortest operating time. However, in greater depths, this variation decreases. This is probably caused by the dark colour of Tašovice rock surface, with lower albedo. Greater in-depth temperature variation is present at Pastýřská rock site. However, these data can be biased by different time-series lengths. Overall differences caused mainly by lithology and aspect are visible.



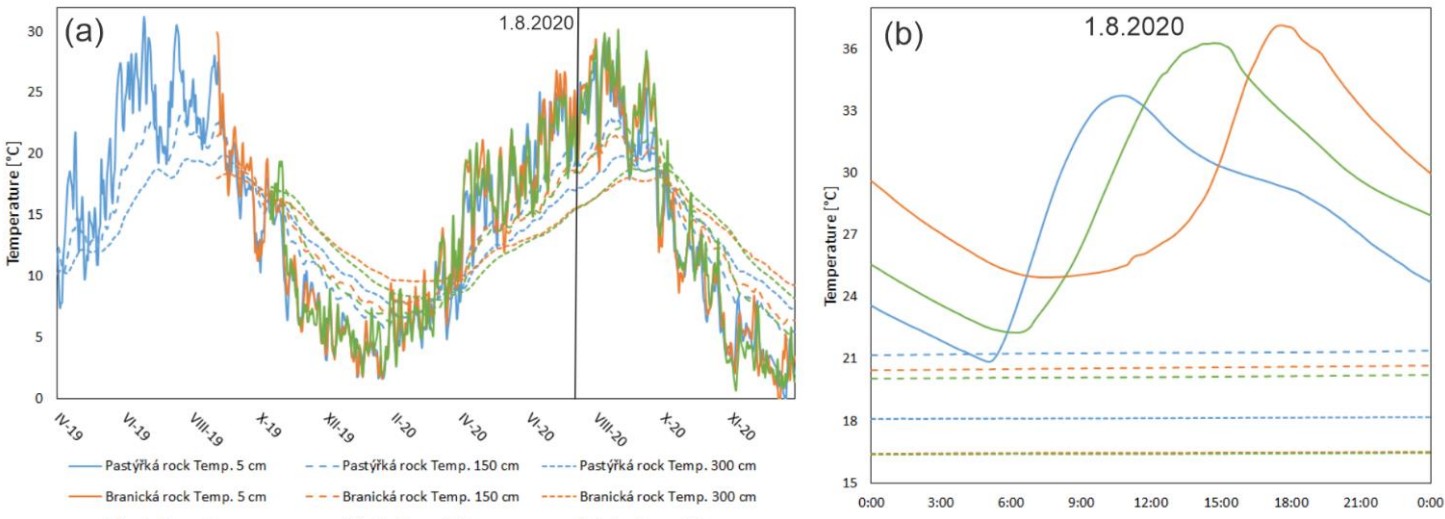

**Figure 5: Comparison of temperatures in different rock slope depths (5, 150 and 300 cm) at three monitored rock slopes. (a): long-term data (daily average), (b): one day data from 1.8. 2020.**

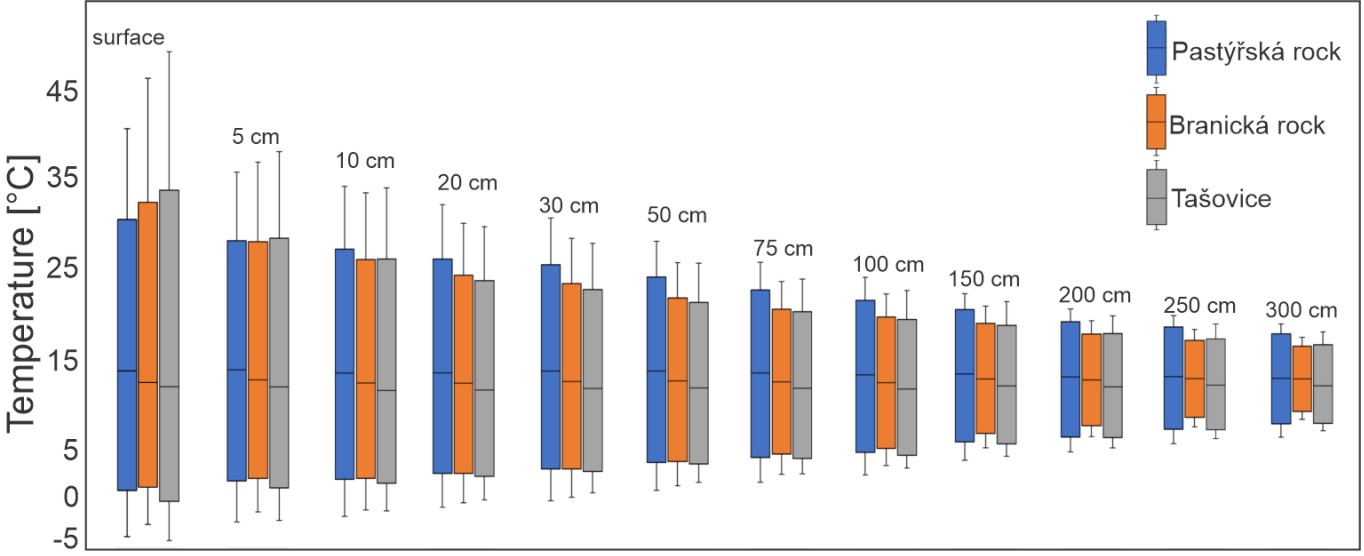

**Figure 6: Comparison of in – depth rock mass temperature data from all three monitored sites.**

**5.4 Blocks dilatation**

At all monitored sites, we are observing the thermally-induced dilatation of individual blocks, however, due to relatively short time-series, the measured crack movements do not show significant opening or closing trends yet. From the measured dilatation data, diurnal and annual amplitudes of crack opening for each instrumented block can be identified. Fig. 7 shows measured diurnal and annual rock block dilatation at Pastýřská rock site. From the figure it is visible the influence of diurnal and annual

temperature changes on the crack opening. Similar behaviour is observed on all monitored blocks (Table 6). The amplitude of crack meters position differs between individual sites and blocks. These differences are caused by different blocks dimensions, crack meters placement and the regime of destabilization.

| Site | Block | Crack meter position amplitude Δ l [mm] | | | | measuring since |
| | | CM1-P1 | CM1-P2 | CM2-P1 | CM2-P2 | |
|---|---|---|---|---|---|---|
| Pastýřská rock | 1 | 1.05 | 0.95 | 0.75 | 0.75 | 23.10.18 |
| Branická rock | 1 | 1.45 | 0.35 | 0.25 | N/A | 4.6.19 |
| | 2 | 0.4 | 0.5 | N/A | N/A | 20.6.19 |
| | 3 | 0.75 | 0.7 | N/A | N/A | 10.7.20 |
| Tašovice | 1 | 0.65 | 0.25 | N/A | N/A | 4.12.18 |
| | 2 | 0.6 | 0.75 | N/A | N/A | 4.12.18 |
| | 3 | 0.85 | 0.7 | N/A | N/A | 18.10.19 |

**Table 6: Amplitude of crack meter measuring at Pastýřká rock: 1 block 4 crack meters, Branická rock: 3 blocks 7 crack meters and Tašovice: 3 blocks 6 crack meters. The table shows the difference between maximal and minimal opening of all placed crack meters. Last measured data: 27.1.2021.**

So far, relatively high crack meter amplitudes were measured on Block 1 (aprox. 170 m$^3$) at Pastýřská rock site and on Block 1 (aprox. 16 m$^3$) at Branická rock site. These blocks are the two largest ones instrumented. Measured crack meter amplitude is caused by block thermal expansion/contraction. On the other hand, relatively small block 3 (BS site) shows relatively large movements although is instrumented only since summer 2020. These movements points on possible gradual destabilization of this block. Blocks that are instrumented at Tašovice site seems to be more stable. Only Block 3 shows 0.85 mm of movement. Again, this block was instrumented recently at the end of 2019. By further monitoring, trend analyses should reveal possible blocks´ destabilization trends. Larger blocks (PS1, BS1; BS2) shows the largest overall amplitude of movements. Rest of smaller blocks shows smaller overall amplitudes, however these seams to be more influenced by the short-term diurnal temperature changes. Sensitivity to fast heating/cooling, makes these blocks more susceptible to temperature-induced irreversible movements.





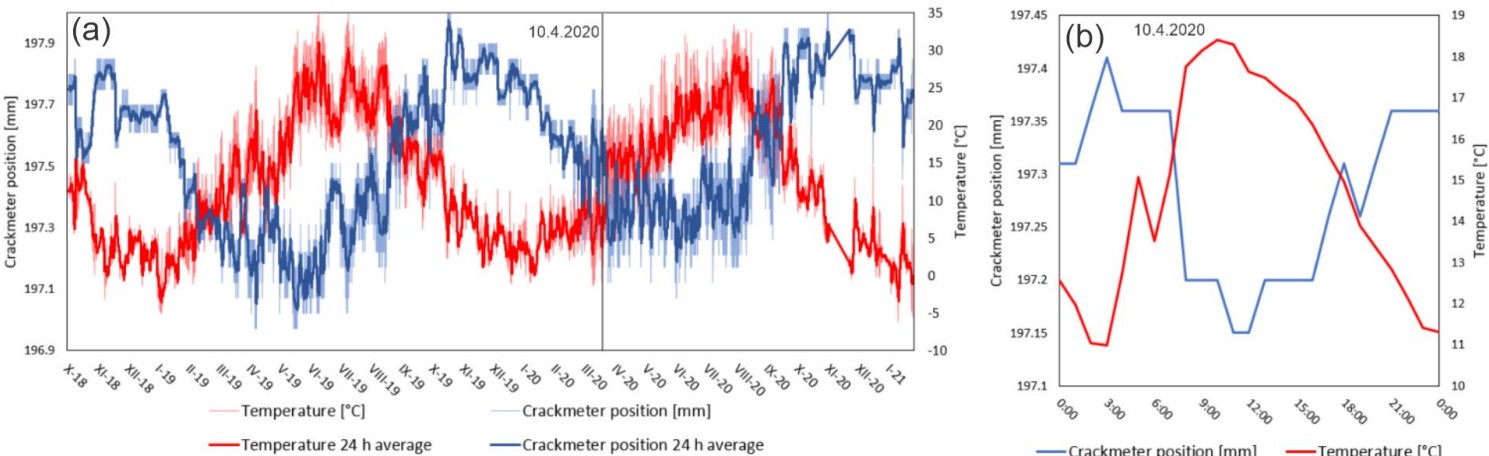

**Figure 7: Measured in situ temperature and crack opening at Pastýřská rock site.(a): whole time-series with annual amplitude, (b): example of diurnal amplitude measured on 10.4.2020.**

## 6 Discussion

Commonly used rock stability monitoring systems are often designed to provide an early warning (Jaboyedoff et al.,
2011; Crosta et al., 2017), aiming primarily at the identification of a hazard and not to investigate the causes or thresholds of the movement acceleration. The presented complex monitoring system is designed to contribute to explaining the various influences on the destabilizing processes, which leads to the eventual loss of rock mass stability and rock fall event triggering. Fantini et al., (2017) have concluded that it is the temperature variations (rather than precipitation or wind) that cause changes in strain within the rock mass leading to its destabilization. However, to assess the strain changes within the mass, it is
necessary to have information on the temperature distribution inside the rock. This is the crucial advantage of the presented monitoring system, as the borehole temperature monitoring allows to identify short and long-term temperature changes up to 3 m depth.

To observe individual influences of the strain in the rock masses, we have placed the monitoring on rock slopes with various aspect (different insolation and its diurnal and annual changes) and built of different rocks (sandstone, granite and
limestone) to include the influence of heat conductivity, capacity and colour of the rock. While there are numerous laboratory studies on rock conductivity (cf. Blásquez et al., 2017), modelling of heat flow based on surface observation (Hall and André, 2001, Marmoni et al., 2020), or coarse, large-scale experiments usually aiming at heat management in the thermal energy industry (Zhang et al., 2018), only a few experiments have been carried concerning the shallow, first meters surface of the rocks (Greif et al., 2017), even though this is the most strained and weathered part of the rock mass (Marmoni et al., 2020).
Moreover, thermal conductivity can be spatially determined from heating/cooling rates of rock slope surface using thermal camera (Pappalardo et al., 2016; Pappalardo and D'Olivo, 2019; Fiorucci et al., 2018; Guerin et al., 2019).



The analyses of structural properties of rock were performed using traditional field compass measurements and using automatic discontinuity extraction from the point clouds. While generally, the results were similar, the point cloud analysis does not include discontinuity sets that are not forming the surface of the rock face. This effect is visible mainly in the case of

the Tašovice 3D model, where the structural setting is not so straight forward as it is at Branická rock and Pastýřská rock sites formed by sedimentary layers.

Concerning the proposed monitoring system, it is compact, built of cheap and easily accessible off-the-shelf components, and easy to modify according to specific conditions at rock the slope site. The performance of the monitoring system was so far without major problems. One crack meter datalogger was damaged and one pyranometer was destroyed by a rockfall triggered

by a severe thunderstorm. Otherwise, monitoring works reliably at all instrumented sites. Maintenance is consisting of changing datalogger batteries and cleaning rain gauge buckets. Online data transfer via Sigfox IoT network (dilatometers) and GSM (environmental stations) works without problems.

| Method | Results | Range | Precision | Sampling rate | Online data | Price |
|---|---|---|---|---|---|---|
| Induction crack meter | 1D distance | < 1 m | 0.01 mm | seconds-days | yes | 300 € |
| Precision tape | 1D distance | < 30 m | 0.5 mm/30 m | hours-days | no | 800 € |
| Fixed wire extensometer | 1D distance | 10 - 80 m | 0.3 mm/30 m | hours-days | yes | 4 000 € |
| Rod for crack opening | 1D distance | < 5 m | 0.5 mm | hours-days | no | 300 € |
| LVDT | 1D distance | < 0.5 m | 0.25 mm | seconds-days | yes | 170 € |
| Laser dist. meters | 1D distance | < 1000 m | 0.3 mm | seconds-days | yes | 1 500 € |
| Portable rod dilatometer | 1D distance | < 1 m | 0.1 mm | hours-days | no | 350 € |
| Total station triangulation | 3D distance | < 1000 m | 5 - 10 mm | hours-days | yes | 3 000 € |
| Precise levelling | 1D distance | < 50 m | < 1 mm | days | no | 350 € |
| EDM | 1D distance | 1 - 15 km | 1 - 5 mm | minutes - days | no | 10 000 € |
| Terestrial photog. | 3D distance | < 100 m | < 20 mm | hours-days | yes | 1 000 € |
| Aerial photog. | 3D distance | < 100 m | 10 - 100 mm | days | no | 1 500 € |
| Tiltmeter | inclination change | ±10° | 0.01° | seconds-days | yes | 300 € |
| GPS | 3D distance | Variable | < 5 mm | seconds-days | yes | 2 000 € |
| TLS | 3D distance | Variable | 5 - 100 mm | hours-days | yes | 100 000 € |
| GB InSAR | 3D distance | Variable | < 0.5 mm | hours-days | yes | 100 000 € |

**Table 7: A comparison of rock slope spatial change monitoring techniques (updated after Klimeš et al., 2012).**

Crack meters can record movements smaller than 0.1 mm (Tables 1,7). In comparison with other methods that measure spatial change, their precision is high, with lower costs (Table 7). The temporal resolution of the measurement is nearly continuous when the crack meter position can be read every second (Table 7). Moreover, we have tested these in a controlled temperature environment using a climate chamber to find out any temperature-dependent errors. In this controlled test, we were able to measure the expansion of a concrete block. The resulting block expansion measurements matched

theoretically calculated concrete block expansion. This way we made sure, that measurement of the crack meters is not biased





by dilatation of the device itself. A disadvantage of crack meter use is that this method provides only one-dimensional spatial change data. To get full 3D data about an unstable feature´s spatio-temporal behaviour, more crack meters must be deployed. Also, the maximum range of this device is limited to 200 mm. That limits the use of this crack meters to changes with lower magnitude.

| Crack meter | PR1_1 | PR1_2 | PR2_1 | PR2_2 | BR1_1 | BR1_2 | BR2_1 | BR3_1 | BR3_2 | BR4_1 | BR4_2 | T1_1 | T1_2 | T2_1 | T2_2 | T3_1 | T3_2 |
|---|---|---|---|---|---|---|---|---|---|---|---|---|---|---|---|---|---|
| **Mean** | 197.55 | 99.18 | 100.36 | 57.34 | 109.10 | 131.13 | 108.21 | 80.77 | 21.56 | 53.61 | 73.23 | 62.64 | 90.53 | 130.65 | 125.39 | 115.86 | 112.41 |
| **Variance** | 0.05 | 0.03 | 0.02 | 0.01 | 0.14 | 0.00 | 0.00 | 0.01 | 0.01 | 0.02 | 0.01 | 0.01 | 0.00 | 0.03 | 0.76 | 0.03 | 0.02 |
| **Stdev** | 0.23 | 0.17 | 0.13 | 0.12 | 0.38 | 0.06 | 0.06 | 0.10 | 0.09 | 0.14 | 0.10 | 0.11 | 0.04 | 0.16 | 0.87 | 0.17 | 0.14 |
| **Median** | 197.56 | 99.19 | 100.36 | 57.34 | 109.06 | 131.14 | 108.23 | 80.78 | 21.59 | 53.63 | 73.26 | 62.66 | 90.55 | 130.74 | 125.96 | 115.80 | 112.43 |
| **Q3** | 197.75 | 99.34 | 100.46 | 57.44 | 109.50 | 131.14 | 108.28 | 80.83 | 21.64 | 53.68 | 73.31 | 62.71 | 90.55 | 130.79 | 126.01 | 116.00 | 112.53 |
| **Q1** | 197.36 | 99.05 | 100.27 | 57.24 | 108.77 | 131.09 | 108.18 | 80.68 | 21.49 | 53.53 | 73.16 | 62.56 | 90.50 | 130.50 | 125.42 | 115.75 | 112.28 |
| **Min** | 196.95 | 98.60 | 100.02 | 56.90 | 108.28 | 130.94 | 108.03 | 80.54 | 21.29 | 53.14 | 72.92 | 62.27 | 90.40 | 130.31 | 122.64 | 115.46 | 107.45 |
| **Max** | 198.00 | 99.59 | 100.75 | 57.63 | 109.74 | 131.33 | 108.38 | 80.98 | 21.78 | 53.97 | 73.60 | 63.10 | 90.70 | 130.89 | 126.69 | 116.29 | 112.67 |


**Table 8: Basic statistical descriptions of data measured by all 17 installed crack meters. PR: Pastýřská rock, BR: Branická rock, T: Tašovice**

In the case of environmental monitoring, we have found differences between sites (Table 4, 9), caused by aspect and local microclimate. Some differences between sites are caused by different length of meteorological variables time-series

(Table 4). When temperature data from in-depth monitoring are compared, differences between monitored sites are apparent (Figures 5,6), both diurnal and annual temperature cycles, and as deep as 3 m. These differences are caused by the combination of the different orientation of rock slopes and by the thermal behaviour of the different rock types. As concerns the energy supply, the solar panel is capable of keeping the battery charged even during cloudy weather or snowy winters. In case of in-depth temperatures, highest differences are observed in surface zone (Table 9). In 3 m depth, is at all sites temperature approx.

same.

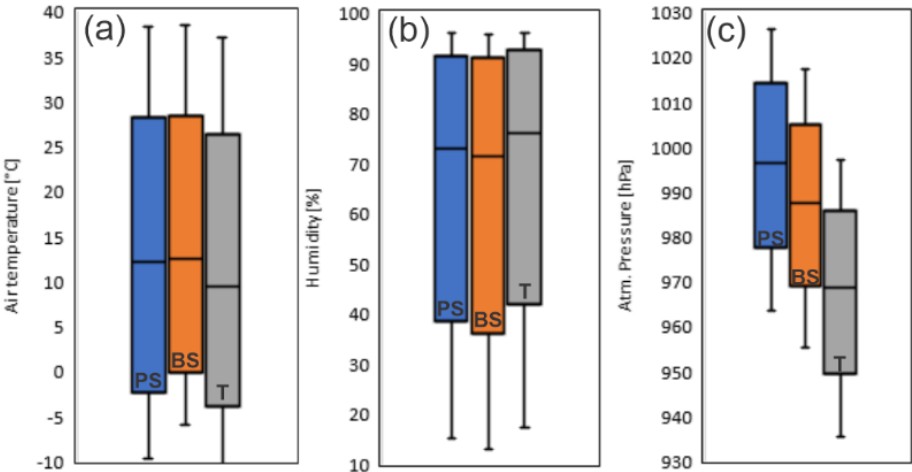

**Figure 8: Comparison of air temperature (a), humidity (b) and air pressure (c) data sets, between Pastýřská rock (PS), Branická rock (BS) and Tašovice (T) sites.**





Solar radiation balance is not directly comparable, due to different aspect and slope of monitored rock slabs. However,
the temporal shift in maximum radiance caused by rock slope aspect is visible from resulted radiation data (Fig. 4). When
whole year data about solar radiance will be available in spring 2021, more differences should be found. Then the comparison
of long-term solar radiation cycles will be possible.

| | T Air [°C] | | | 1 h. prec. [mm] | | | Humidity [%] | | | Air pressure | | | T rock face [°C] | | | T 5 cm [°C] | | | T 10 cm [°C] | | | T 20 cm [°C] | | |
|---|---|---|---|---|---|---|---|---|---|---|---|---|---|---|---|---|---|---|---|---|---|---|---|
| | PR | BR | T | PR | BR | T | PR | BR | T | PR | BR | T | PR | BR | T | PR | BR | T | PR | BR | T | PR | BR | T |
| **Mean** | 12.1 | 12.4 | 9.4 | 0.1 | 0.1 | 0.0 | 72.9 | 71.2 | 75.8 | 996.4 | 987.4 | 968.5 | 14.6 | 13.3 | 12.8 | 14.5 | 13.4 | 12.6 | 14.6 | 13.5 | 12.7 | 14.7 | 13.5 | 12.7 |
| **Variance** | 72.0 | 71.5 | 71.6 | 0.3 | 0.5 | 0.1 | 301.6 | 324.9 | 305.1 | 75.2 | 72.8 | 79.0 | 79.6 | 80.9 | 98.3 | 67.9 | 65.9 | 72.9 | 65.4 | 61.7 | 65.3 | 61.1 | 55.6 | 57.8 |
| **Stdev** | 8.5 | 8.5 | 8.5 | 0.5 | 0.7 | 0.2 | 17.4 | 18.0 | 17.5 | 8.7 | 8.5 | 8.9 | 8.9 | 9.0 | 9.9 | 8.2 | 8.1 | 8.5 | 8.1 | 7.9 | 8.1 | 7.8 | 7.5 | 7.6 |
| **Median** | 11.4 | 11.9 | 8.2 | 0.0 | 0.0 | 0.0 | 78.8 | 76.6 | 82.6 | 996.6 | 987.7 | 968.9 | 14.1 | 11.7 | 10.5 | 14.5 | 12.3 | 11.0 | 14.7 | 12.5 | 11.3 | 14.9 | 12.8 | 11.6 |
| **Q₃** | 18.3 | 18.5 | 15.5 | 0.0 | 0.0 | 0.0 | 86.8 | 86.1 | 89.3 | 1002.0 | 992.9 | 974.3 | 21.3 | 19.4 | 19.2 | 21.1 | 19.8 | 19.4 | 21.3 | 19.8 | 19.4 | 21.3 | 19.7 | 19.0 |
| **Q₁** | 4.9 | 5.5 | 2.4 | 0.0 | 0.0 | 0.0 | 61.6 | 58.9 | 66.5 | 991.3 | 982.7 | 963.3 | 6.5 | 5.8 | 4.5 | 6.8 | 6.3 | 5.2 | 7.0 | 6.5 | 5.4 | 7.2 | 6.7 | 5.7 |
| **Min** | -9.8 | -5.9 | -10.4 | 0.0 | 0.0 | 0.0 | 15.1 | 12.9 | 17.4 | 963.6 | 955.3 | 935.5 | -3.7 | -2.4 | -4.2 | -2.1 | -1.1 | -2.0 | -1.3 | -0.6 | -0.7 | -0.3 | 0.1 | 0.5 |
| **Max** | 38.2 | 38.4 | 37.0 | 28.6 | 36.5 | 18.8 | 96.1 | 95.8 | 96.1 | 1026.3 | 1017.2 | 997.1 | 41.3 | 46.9 | 49.8 | 36.2 | 37.2 | 38.4 | 35.4 | 34.6 | 35.2 | 33.5 | 31.4 | 31.0 |

| | T 30 cm [°C] | | | T 50 cm [°C] | | | T 75 cm [°C] | | | T 100 cm [°C] | | | T 150 cm [°C] | | | T 200 cm [°C] | | | T 250 cm [°C] | | | T 300 cm [°C] | | |
|---|---|---|---|---|---|---|---|---|---|---|---|---|---|---|---|---|---|---|---|---|---|---|---|---|
| | PR | BR | T | PR | BR | T | PR | BR | T | PR | BR | T | PR | BR | T | PR | BR | T | PR | BR | T | PR | BR | T |
| **Mean** | 14.7 | 13.5 | 12.7 | 14.6 | 13.5 | 12.8 | 14.5 | 13.5 | 12.8 | 14.4 | 13.6 | 12.9 | 14.3 | 13.7 | 13.0 | 14.1 | 13.8 | 13.0 | 14.0 | 13.8 | 13.1 | 13.9 | 13.8 | 13.1 |
| **Variance** | 56.8 | 50.4 | 52.9 | 49.7 | 41.9 | 46.0 | 43.3 | 34.4 | 39.4 | 37.7 | 29.0 | 34.4 | 29.2 | 21.0 | 26.3 | 23.3 | 15.2 | 20.2 | 19.0 | 11.4 | 15.7 | 15.6 | 8.6 | 12.3 |
| **Stdev** | 7.5 | 7.1 | 7.3 | 7.0 | 6.5 | 6.8 | 6.6 | 5.9 | 6.3 | 6.1 | 5.4 | 5.9 | 5.4 | 4.6 | 5.1 | 4.8 | 3.9 | 4.5 | 4.4 | 3.4 | 4.0 | 3.9 | 2.9 | 3.5 |
| **Median** | 15.1 | 13.0 | 11.9 | 15.1 | 13.4 | 12.4 | 14.9 | 13.7 | 12.9 | 14.7 | 13.7 | 13.2 | 14.2 | 14.0 | 13.0 | 13.8 | 13.7 | 13.0 | 13.9 | 13.7 | 12.9 | 13.8 | 13.7 | 12.9 |
| **Q₃** | 21.2 | 19.3 | 18.5 | 21.0 | 18.8 | 17.9 | 20.6 | 18.6 | 17.6 | 20.2 | 18.3 | 17.4 | 19.6 | 18.0 | 17.0 | 18.8 | 17.4 | 16.9 | 18.2 | 16.8 | 16.5 | 17.7 | 16.5 | 16.2 |
| **Q₁** | 7.4 | 7.0 | 6.0 | 7.6 | 7.4 | 6.4 | 7.8 | 7.8 | 6.7 | 8.2 | 8.2 | 6.9 | 8.8 | 9.3 | 7.9 | 9.2 | 9.9 | 8.6 | 9.8 | 10.6 | 9.2 | 10.3 | 11.1 | 9.8 |
| **Min** | 0.3 | 0.7 | 1.3 | 1.5 | 2.0 | 2.3 | 2.4 | 3.3 | 3.3 | 3.3 | 4.3 | 4.0 | 4.7 | 6.1 | 5.2 | 5.8 | 7.5 | 6.2 | 6.6 | 8.5 | 7.2 | 7.4 | 9.4 | 8.1 |
| **Max** | 31.6 | 29.3 | 28.8 | 28.9 | 26.6 | 26.5 | 26.8 | 24.6 | 24.9 | 25.4 | 23.5 | 23.9 | 23.2 | 21.8 | 22.3 | 21.7 | 20.3 | 20.9 | 20.7 | 19.2 | 19.8 | 19.9 | 18.4 | 19.0 |

**Table 9: Basic statistical descriptions of atmospheric and borehole rock mass temperature monitoring. PR: Pastýřská rock, BR:**
**Branická rock, T: Tašovice.**

It is necessary to remark that the destabilisation processes are rather slow and have a low magnitude in the central
European mid-latitude climate. Therefore, long-term time series monitoring is necessary. Also, there are several cycles with
different length, amplitude and depth-reach, ranging from diurnal cycles up to long-term cycles linked with solar activity or
climatic oscillations (Gunzburger et al., 2005; Sass and Oberlechner., 2012; Pratt et al., 2019). Among these are the most
prominent diurnal and annual cycles (Marmoni et al., 2020). Diurnal cycles have shallower reach (Fig. 5), but are fast and thus
cause high strain in the surficial rock layer, while annual cycles are slower, but with higher amplitudes and depth reach (Hall
and André, 2001). This information helps to clarify the role of thermally-induced stress in rock disintegration. Also, in
combination with the temperature and global radiation measurements, heat conduction velocity inside rock mass can be
determined. Diurnal temperature cycles with higher magnitude can play a crucial role in rock fall triggering. This, together
with mechanical properties of the rock mass (Table 2), will allow creating more accurate thermomechanical models of the
monitored rocks slope in the future. These models will be used to identify zones where the accumulation of thermally induced
stress concentrates, as the places of potential destruction and following destabilization of the rock slope. On all sites, the
highest diurnal measured crack meter movements are recorded in the spring and autumn months, when diurnal rock slope





surface temperature changes have the largest magnitude. These conditions when the temperature at night falls 0°C and during
daytime again rises, are crucial to freeze-thaw cycles development and consequent destabilization of the rocks.

In other works that using similar instrumentation was published in past. (Matsuoka, 2008; Bakun-Mazor et al.,
2013,2020; Dreabing, 2020; Draebing et al., 2017 Nishi and Matsuaoka, 2010), although in thermally induced rock mass
deformations monitoring is still relatively marginally studied. Matsuoka (2008) presented long-term data of crack meter
monitoring rock slope unstable parts in alpine environment. Same as in our results, measured joint dynamic is influenced by
air and rock mass temperature. Similarly, to our first data, dynamic of monitored joints is highest in spring and autumn. Because
longer time span of monitoring Matsuoka (2008) measured gradual, temperature driven joint opening. Most significant joint
opening is in his work linked with freeze-thaw conditions in alpine environment. Nevertheless, even in dynamic alpine
environment, joint opening is slow, spanning approx. 0.4 mm in 2 years of monitoring. Is expected, that in temperate climate
these processes are even slower. Nishi and Matsuaoka (2008) presented influence of temperature to large rock slide
displacement. In this, to our sites different setup, they have measured large displacement over 1 m in 3 years of monitoring.
Largest movements velocities were documented during highest precipitations seasons. Due to different spatial extent of
monitored rock mass movements are results almost incomparable. Bakun-Mazor et al., (2013, 2020) proposed monitoring
system to distinguish thermally and seismically induced joint movements in limestone and dolomites a Masada cultural
heritage site. In this work amplitude of thermally induced joint movements was approximately 0.3 mm in one year. Which is
similar to our first results. In this work, they have estimated annually irreversible joint opening about 0.2 mm. However, in
this study, thermally induced movements are supplemented with seismically induced movements with higher magnitude. We
hope, that in long-term, we will be able to observe similar wedging-ratcheting mechanism at our sites, where also effect of
frost shattering should play not negligible role. Draebing et al., (2017) and Draebing (2020) observed crack opening in alpine
environment. In this extreme environment, they were able to observe short-term ice wedging induced movements up to 1 mm
in several days. These movements were active in snowmelt season, when ice wedging is most active. By comparing in situ
crack meter temperature and crack meter opening they have established linkage between in situ temperature and joint dynamic.
In this case joint dynamic was also influenced by snow cover, which has in alpine region longer time-span than in case of our
monitoring sites. However, even in these conditions, gradual irreversible joint opening is relatively slow, about 0.1 mm/year.
We hope, that data from winter 2020/21 bring similar results in case of our monitoring, however with lack of active permafrost
and ice filled joints at our sites, these moments should have lower magnitude. Newly instrumented site in Krkonoše mountains
should provide data from dynamic mountainous region.

Measuring temperature inside rock mass is nowadays relatively uncommon technique. Only few works estimate in
depth rock mass temperatures in surface zone (Magnin et al., 2015a; Fantini et al., 2018). In work of Magnin et al., (2015a) is
measured rock mass temperature inside 10 m deep borehole. Boreholes were drilled in alpine, permafrost active areas and this
research is oriented mainly to estimate active permafrost depth ant its temporal evolution. In shallow surface zone, they have
recorded annual temperature differences approx. 5°C in 3 m depth. Temperature amplitude is rising in shallower subsurface
zones. Our data from horizontal boreholes shows amplitude in 3 m approx. 10° C. This is caused by warmer climate and



absence of long-term snow cover on rock face. Fantini et al., (2018) studied short-term temperature profiles in experimental
limestone rock slope. Diurnal temperature cycles reached maximum depth of approx. 20-30 cm. These results correspond with
our measurements, where we are able to observe diurnal temperature cycles up to 50 cm depth, during summer, when rock
mass surface is intensively heated by solar radiation. Is necessary to mention, that is not easy to compare results, with these
works, due to different climatic setup.

Currently, the three sites are continuously measuring for a period between 1 and 2 years (Table 4). Based on this, we
can show that the system is capable of observing the influence of the response of the monitored blocks on the thermal dilatation
(Fig. 6). However, to exclude seasonality, the time-series of the crack opening should be longer than 2-3 years. In a longer
period, we expect to observe the process of long-term rock slope destabilization represented by a gradual trend of crack
opening/closure, which points on to the partial block destabilization. Longer time series also allows users to observe seasonal
statistical trend tests to describe the influence of meteorological variables on the rock blocks stability.

## 7. Conclusions

A newly designed complex rock slope stability monitoring system was introduced. The presented complex monitoring
system combines monitoring of meteorological variables with 3 m deep in-rock thermal profile and dilatation of the unstable
rock block joints. It brings a unique opportunity to observe long-term gradual changes within the rock mass, destabilizing the
rock slope.

The design of the system allows an easy installation at various locations without major adjustments or changes. All
components of the system are available off-the-shelf, at a relatively low price and are easy to replace with low skill
requirements. The environmental data are transferred via GSM to a remote server, and the dilatation data are sent via IoT
SigFox network or can be downloaded remotely from several tens of meters. Thus, the maintenance visits of the sites can be
limited to several months' interval.

The monitored sites are easily comparable as similar monitoring set-up and equipment is used. Thus, we are
monitoring the reaction of various rock types on a certain climatic event and observing the differences and similarities on
particular sites. This concerns not only movements or expansion of the rock mass but also the heat advance into the rock, its
velocity and amplitudes, otherwise very difficult to measure. Significant differences in shallow surface rock mass zone are
observable from 3 m borehole thermometer data.

Further development of this project should include the installation of in-situ rock stress monitoring. Measuring of joint
movements combined with temperature and other external influencing factors will be analysed to understand mechanisms of
individual processes, leading to rock slope destabilization. Moreover, it can contribute to explaining the influence of the
individual destabilizing processes. Local factors which influence the rock slope stability are described using classical and
modern geomorphological, geomechanical or remote sensing methods. Structural and laboratory-measured mechanical rock
properties will be used for heat flow and heat-induced strain numerical modelling within the rock mass.






## Data availability

Data available: https://data.mendeley.com/datasets/4t38tvb4yn/draft?a=f9020d9b-fbd3-4489-a1ca-0e4ffd623212

## Authors contribution

O. Racek and J. Blahůt designed system and directed instrumentation of sites and continuously processing data and maintain

monitored sites

O. Racek processed crack meters data

J. Blahut processed in depth temperature data and environmental data

F. Hartvich supervised all works, helped with graphic parts of manuscript

## Competing interests

"The authors declare that they have no conflict of interest."

## 8 Acknowledgements

This research was carried out in the framework of the long-term conceptual development research organisation RVO: 67985891, TAČR project number SS02030023 "Rock environment and resources" within program "Environment for life", internal financing from Charles University Progress Q44 and SVV (SVV260438) and the Charles University Grant Agency

[GAUK 359421]

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
