# Peer review of "Observation of the rock slope thermal regime, coupled with crack meter stability monitoring"

_Geoscientific Instrumentation, Methods and Data Systems, 2021_

## Author Response (AR1)

**Dear authors and editor,**

**This paper presents a setup to monitor crack opening changes due to temperature variations in rock faces. It includes the following sensors: weather station and pyranometers, crackmeters and thermometers in boreholes. Three sites have been equipped on three different type of rocks and preliminary results are presented.**

**Monitoring thermal effects on rock slopes stability is a relatively new type of investigation and it is particularly interesting to understand the long term weakening of rock masses that eventually leads to failure. In my point of view this paper makes two original contributions. First it describes at setup that is robust enough for long term monitoring (several years) with a minimum of maintenance. Second a new temperature logger for boreholes was apparently designed or assembled from available pieces, however this device is not clearly described.**

**In my opinion, the weakness of this setup is that all temperatures are air temperatures. There is no direct measurement of rock temperature by contact thermometers (thermoresistance or thermocouple sealed to the rock). For instance, during a sunny summer afternoon, the rock temperature is quite often 10 to 20°C higher than the air temperature (max air and rock temperatures can also be shifted in time). Line 246 states that air temperature influences the dilatation of the blocks – this is only partly true. Correlation does not imply causation. In summer solar radiations will heat the rock mass, that will heat the surrounding air. A contact thermometer should be added to get a reliable rock surface temperature.**

Answer: From previous version of manuscript wasn't clearly explained system of borehole temperature probe and direct rock surface temperature measurement. Newly added description of this device, together with scheme, should explain this part of monitoring better. Our borehole temperature probe measures directly temperature of surrounding rock mass and additionally there is one thermocouple placed directly on rock slope surface. In case of influence of air temperature to block dilatation, air temperature is responsible mainly for longer annual dilatation cycles, where long term rock mass temperature is function of annual air temperature course.

Changes in manuscript: Description of borehole temperature probe was added. In marked-up manuscript line 240. Also new figure to better understanding was added: Fig. 2.

**The air temperatures measured in the borehole can be at equilibrium with the local rock temperatures if the sections are perfectly sealed by the insulating material. But the paragraph §2.3 is not very informative about this part of the setup. As this is the most innovative contribution, that would be nice to have a picture of it and some explanations how it is introduced in the borehole.**

Answer: Thank you for this suggestion. Part about compound borehole temperature probe was enlarged and new picture with scheme of device was added. Changes in manuscript are highlited.

*Corrections and suggestions to authors*

**The abstract should be written again. It is too general and looks more like an introduction / advertisement. Here it emphasizes the innovative aspect of the setup, but at the end we still don't know what is new, specifications of sensors, etc.**

Answer: Abstract was rewritten according these demands. Hopefully, now it represents whole article in better way.

Changes in manuscript: Abstract was rewritten. Changes visible at line 24.

**Different terms are used to refer to "weather stations" (environmental station, etc..). I would keep "weather stations" for the whole paper.**

Answer: Thanks for suggestion. It was our mistake and inattention. In second version of manuscript we maintain continuity with "weather station" therm.

Changes in manuscript: Term was changed to "weather station" in whole manuscript.

**Table 1 and rest of the paper: I guess that all the W/m3 should be W/m2**

Answer: Our mistake. We have rewritten tables and concerned parts of manuscript.

Changes in manuscript: Tables description and text was changed. Line 165, 224

**Table 2: add definition of symbols in legend. Why do the two unweathered sandstones have so different properties? the 1st one is odd.**

Answer: Legend was rewritten. Table was reworked to be clearer. Second unweathered line actually belongs to Branická rock limestone values.

Changes in manuscript: Changes visible at line 305.

**Table 3: some mistakes in measurement reporting (180/30). Use 3 digits for dip direction (0xx)**

Answer: Thank you for your suggestion, it was reworked according your comment.

Changes in manuscript: Changed table at line 325

**Table 5: is the pyranometer measuring every 10 min too?  I don't see the point of this table, it can be supressed. Most of data are zero because of the night.**

Answer: Yes, pyranometer measures every 10 minutes, same as all environmental and borehole temperature monitoring. Following your suggestion table was supressed.

Changes in manuscript: Table supressed

**Figure 5: on the version provided I cannot see the line the corresponding to Branicka rock temp at 300 cm on (b)**

Answer: Graph was reworked to be readable clearly.

Changes in manuscript: Reworked graph is at line 390.

**Figure 7: cannot read the lines corresponding to temperatures (light grey)**

Answer: Graph was reworked. Hopefully in second version manuscript it is easier to read.

Changes in manuscript: graph at line 440.

**Figure 8: I don't see the interest of this figure for the present contribution. Suppress**

Answer: Figure 8 was supressed.

Changes in manuscript: Figure supressed

**L55: English proofreading / rewriting**

Answer: rewritten

Changes in manuscript: Whole introduction part was reworked.

**L70: Chen 2017 missing**

Answer: Chen 2017 citation was added to literature.

Changes in manuscript: citation added at line 713

**L91: delete ","**

Answer: deleted

Changes in manuscript: deleted

**L100 and L135: can you define the "global radiation balance"? what is global? where is the balance?**

Answer: Actually, better expression is in this case solar radiation balance of rock slope surface (incoming/reflected solar radiation). Global radiation balance concerns whole planet and is not limited only to solar radiation. Thank you for your suggestion, in new version of manuscript is this sentence rewritten.

Changes in manuscript: rewritten to solar radiation. Line: 94, 131, 154, 214

**L147: the wavelength is certainly 2800 nm, not 1200 nm**

Answer: Yes, sorry for this mistake. It was rewritten in new version of manuscript.

Changes in manuscript: rewritten at line 224

**L170: English proofreading / rewriting**

Answer: Rewritten

Changes in manuscript: line 260

**L281-283: English proofreading / rewriting**

Answer: Rewritten

Changes in manuscript: Line 395

**L354: English proofreading / rewriting**

Answer: Rewritten

Changes in manuscript: Line 500

**L381-417: English proofreading the whole paragraph + errors in the references**

Answer: Whole section was reworked.

Changes in manuscript: Rewritten discussion starts at line 44**2**

**Anonymous referee 2**

**The paper entitled « Observation of the rock slope thermal regime, coupled with crack meter stability monitoring » presents a monitoring system of the thermal and rheological behaviour at the surface and near-surface of three rock wall sites in Czech Republic.**

**The paper is overall well-written and easy to follow, the monitoring approach is interesting but significant improvements are required before publication.**

**My main concern is that the authors claims that the monitoring system is original and innovative because of its completeness and affordability. While this is true that the system is affordable, it is, in my opinion, not unique and not so innovative : many sites are equiped with both ground temperature measurements in shallow boreholes and crackmeters (e.g. Weber et al., 2019, Ewald et al., 2019, Hasler et al., 2012, Gischig et al., 2011, in addition of those already referred). More explanations would be needed to really understand how novel is the system: how (precisely) the data will contribute to improve the understanding of rock fall preparation and triggering that the other system do not allow ? Will it provide data for thermo-mechanical models ? How such improvements can be made ? Which paramaters, which process in the models ? (See more detailed comments).**

Answer: Thank you for your detailed comment. It is true that system itself doesn't use really innovative approach. However, innovation in case of our system comes from use affordable instrumentation on multiple sites, which ensures partial data comparability. Other sites using

similar instrumentation that you mention are typically focused on permafrost degradation monitoring and are located in mountainous alpine region. Our system, on the other hand is placed within mit-latitude region, where thermally forced rock mass destabilization is not linked with permafrost thaw. With similar instrumentation on different sites, we can observe differences between thermomechanical behavior of different rock types, structural setting or aspect according to general directions. Data from crack meters and in-depth temperature observations together with physical and mechanical properties of rock will be later used to construct and validation of thermo-mechanical models. Ongoing innovation of system includes in situ strain monitoring using on surface glued strain gauges.

Changes in manuscript: Manuscript was rewritten according to comment.

**Moreover, many data are presented without being discussed nor used for research perspectives: e.g. Table 2, 3, Figure 3, … This makes the paper quite long without improving its impact. I suggest to rework the paper top make sure that the data that are presented are clearly used for the results, discussion or explanation of research strategy.**

Answer: Thank you, for your suggestions. We have tried to rewrite whole manuscript according to these.

Changes in manuscript: Manuscript was shortened and rewritten.

**Other major comments:**

- **Manuscript content and objectives: they are very vague, specific research questions and a clear research strategy should be cleary explained and detailed.**
  - Answer: New version of manuscript should deal with these demands.

  Answer: Thank you for your comment. Hopefully second version of manuscript is more detailed and provide more information about research strategy and partial results.

- **Title: the title should report at least the study area (Czech Republic, 3 sites) and environnemental settings**

  Answer: Title was revisited. Now is clear that monitoring takes place in Czechia.

- **The abstract needs to be rewritten. It is vague and general. It should be more precise: how many sites intrumented, which environemental settings (elevation range, rock type, etc.), where are these sites, when did the record start, how long are the type series, what type of differences are measured…**
  - Answer: Abstract was rewritten. New version should include all above listed missing parts.
- **Introduction: It is generally quite long, lines 47-74 are a long list of some of the existing instrumentation to monitor and detect rock slope deformation and failure. It is too long and too detailed and I didn't get the purpose of such a long list which is somehow summarized line 73-74 but is not convincing. I am not convinced that approach presented in this pape ris so differentthan many other sites, except maybe that instrumentation is relatively affordable. In the introduction, the hydrological processes are not considered at all while they**

**represent a major external forcing in rock falls (see for example Krautblatter and Moser, 2009). I suggest largely rewritting the introduction to make it more concise and better introduce the approach presneted in the paper in order that the reader understand why it is so different than the others. In my opinion, the quetsion related to the choice of the fractures and blocks to instrument with crackmeters is still open, and many other studies combined such point-scale geotechnical observation with geodesy data to apprehend rock deformation at a larger scale as well, which is after all, a more complete approach than only the geotechnical approach.**

- Hopefully newly rewritten introduction is more clear in care of above mentioned problems. Mentioned list of methods and approaches was shortened and short paragraph mentioning hydrological influence to rock slope stability was added. Of course, point geotechnical data does not represent dynamic of rock slope in its full extent, however other methods that allows that (camera monitoring, TLS, GbSar) are expensive and cannot be placed within multiple sites. Point data about spatial behavior of single crack does not represent whole rock slope, but if unstable block is instrumented this way, influence of exogeneous factors on its stability can be observed. More complex data about whole rock slope can be gained using TLS, SfM photogrammetry or IR camera campaigns. We are planning to perform these campaigns at least one per year for each locality, to find annual changes of rock faces.

- **Section 2.3: the same concern arises. Words such as « complex monitoring », « innovarive » are used but I still do not understand why the approach is so innonvative. Many other studies also combines shallow borehole temperature measurements with crackmeters.**
  - It is true, that instrumentation and approach itself really is not innovative. Therefore, we have changed description of system in this chapter. Main advantage of here presented system is affordability and modularity, so it can be placed within all kinds of rock slopes.

- **It would be interesting for the reader to know if any instrument was calibrated or not.**
  - Answer: All parts of system were calibrated by manufacturer.
  - Changes in manuscript: Calibration of instruments was mentioned in new version of manuscript.

- **Another concern : the instrumentation do provide any data about the temperature and the mechanical behaviour of the failure pla, which, in my opinion, strongly limits the interest to understand failure mechanisms.**
  - Answer: Temperature within failure plane, or discontinuity that defines monitored blocks can be monitored with datalogger placed inside discontinuity. Regime of destabilization of partial blocks was described during field investigation and crackmeters were placed to best capture possible destabilization trends.

- **The paper is long with lot of tables and figures that are not necessarily relevant for understanding the approach. I suggest to create supplementary material or to barely remove information that are not a direct relevance to understand the strategy behind the instrumentation (see detaield comments).**
  - Answer: To address this comment. Irrelevant pictures, tables etc. were removed.
  - Changes in manuscript: manuscript was shortened and irrelevant content was supressed.

Detailed comments :

- **Line 33: permafrost doesn't melt, it thaws.**
  - Answer: it was corrected with right therm.
  - Changes in manuscript: Whole introduction was reworked
- **Line 38: see also the PERMOS reports from the Swiss permafrost monitoring system**.
  - Answer: information about PERMOS system was added in introduction.
  - Changes in manuscript: line 57
- **Line 43: « Unfortunately… » : such formulation is not appropriate in a scientific paper**
  - Answer: sentence was rewritten to meet appropriate formulation.
  - Changes in manuscript: Whole introduction was reworked
- **Line 55-56: there is something wrong with this sentence, rephrase.**
  - Answer: sentence was rewritten.
  - Changes in manuscript: Whole introduction was reworked
- **Line 83: what is a « 2D environment »?**
  - Answer: Sentence meant that temperature is measured both on surface and in rock mass depth. It was misleading formulation and whole paragraph was rewritten.
  - Changes in manuscript: changes starts at line 127
- **Line 95: is teh monitoring system the same for each site?**
  - Answer: Used sensors are same for all monitoring sites. Ofcourse monitored blocks differs in term of dimensions and regime of destabilization. Moreover, different number of crack meters are used within sites.
  - Changes in manuscript: Whole introduction was reworked
- **Table 1: why some items have o price?**
  - Answer: Prices are listed for set. It means one price is for pair of crack meters and datalogger, or for whole weather station (control unit, thermometer, rain gauge etc.).
- **Figure 2: remove « so far » from the caption, it is not appropriate**
  - Answer: Sentence was rewritten.
  - Changes in manuscript: Description of figure was rewritten. Line: 260
- **Table 5: over which period are the radiation measurments available?**
  - Answer: Table was suppressed. Pyranometer data are available since 1/2020 (Branická rock), 2/2020 (Pastýřská rock) and 12/2020 (Tašovice).
  - Changes in manuscript: Table was supressed.
- **Line 268: number of the figure is missing**
  - Answer: Number was added
  - Changes in manuscript: Line 384
- **Line 269: the operating time is not relevant here as all sensors seem to record the greatest amplitude within this period on Fig. 5. If not, clarify.**
  - Answer: You are right, that all sites shows the greatest amplitude within this period. On the other hand at Pastýřská rock site and Branická rock site, time series span over two summer periods with largest surface temperatures fluctuation. This paragraph was rewritten, to clarify, that surface temperature at Tašovice site is caused by lower albedo of dark surface.
  - Changes in manuscript: This part of manuscript was rewritten.

- **Line 271: I do not understand this statement: the temperature amplitude decrease with depth is related to the thermal diffusivity, not the albedo.**
  - Answer: Statement was meant in a way, that lower albedo leads to greater surface temperature amplitude.
  - Changes in manuscript: Changes in manuscript: This part of manuscript was rewritten.
- **Line 272: how sure is this statement ? Please refer your interpretation to facts/proper observations or do not interpret.**
  - Answer: By different aspect is caused temporal shift between diurnal surface temperature peaks. Different lithology causes differences in heat transfer, which cannot be directly observed. Sentence was rewritten.
- **Figure 6: explain teh statistics displayed with the boxplot in the caption: median… ?**
  - Answer: Legend was rewritten.
  - Changes in manuscript: Line 393
- **Line 280: what is a significant opening? Does « significant » have a scientific meaning or definition in this context?**
  - Answer: Opening, that is clearly not caused by thermal expansion of rock mass. It is not statistically defined, so the sentence was rephrased.
  - Changes in manuscript: line 398
- **Line 290: same question with « relatively high »**
  - Answer: Again, sentence was rephrased to describe amplitude larger than 1 mm.
  - Changes in manuscript: line 411
- **Line 293-294: this si very speculative as the time series is very short. Please, base your interpretation on facts rather than speculation. The interpretation might change completly with a longer time series.**
  - Answer: It is speculative, but larger amplitudes over short time signalizes that block is influenced by temperature cycles. That should be confirmed by further monitoring. The sentence was rephrased.
  - Changes in manuscript: line 417
- **Line 295: how does a block destabilization trend is expected to look?**
  - Answer: Trends should be confirmed by statistical analyses. However fast destabilization trend should be visible from graphs, when crackmeter opening/closure is not driven by thermal expansion/dilatation of rock mass.
  - Changes in manuscript: explained at line 430
- **Line 313 ff: the present forcings are not the only factors of rock stability. Past conditions and events (climate change, former rockfall, …) have to be accounted for as well. In addition, the hydrological processes must be considered.**
  - Answer: Thank you for this comment. Forcings, that you mentioned were added to second manuscript.
  - Changes in manuscript: discussion was rewritten according your comments
- **Line 323: the automatic discontinuity extraction approach needs some detailed explanation.**
  - Answer: It was done using freely available DSE software. Citation is listed in manuscript.
  - Changes in manuscript: citation of approach added at line 470
- **Line 335-340: this is part of the method, not the discussion !**
  - Answer: Paragraph was rewritten and moved to methods.
  - Changes in manuscript: line 182

- **Table 7: I also wonder if this is relevant in the discussion. This would be better to explain why the monitoring system of this study is so innovative.**
  - o Answer: Thank you for your suggestion. This table was moved to results.
  - o Changes in manuscript: table moved to line 200
- **Table 8: This is part of the results.**
  - o Answer: Table was suppressed.
  - o Changes in manuscript: table was supressed and new fig 8 was added
- **Figure 8: idem.**
  - o Answer: figure was suppressed
  - o Changes in manuscript: figure supressed
- **Table 9. Idem, and I do not really see the relevance of giving so much details.**
  - o Answer: Table was suppressed.
  - o Changes in manuscript: figure supressed
- **Line 366-367: explain this statement and appropriate references.**
  - o Answer: This is given by shorter period with freeze-thaw activity period, lower solar radiation caused temperature amplitudes or lower overall precipitation. References were added
  - o Changes in manuscript: rewritten at line 519
- **Line 372: explain the concept of « rock disintegration » and how temperature change act for such process.**
  - o . Temperature changes causes irregular heating and cooling of rock mass. These leads to irregularities in rock mass dilatation at surface and in depth, which causes thermally induced stress/strain, which eventually can lead to discontinuity evolution and breakage of rock mass surface layers. Thermally driven disintegration also act in grain scale, where grains of different minerals expand differently and induce stresses in to rock mass (Hall and André, 2001,2003).
  - o Changes in manuscript: Rewritten and reference added at line 528
- **Line 376-377: to determine the place of the potential rock failure, quantitative understanding of the mechanical properties of the failure plan would be required. Similarly, the measured temperature in compact rock is not representative of the temperature in fractures. See Hasler et al., 2011 for example.**
  - o Answer: Numerical modeling will be combined with field data gained by IR camera and radiation data. By our approach places where thermally induced stress is concentrated, will be compared with field data about unstable parts of rock slope.
  - o Changes in manuscript: sentence rewritten at line 308, 356
- **L 390: the sentence is difficult to understand. Please rephrase.**
  - o Answer: Sentence was rephrased.
  - o Changes in manuscript: Line 554
- **Line 412: this statement is ehre again very speculative in the absence of proper comparison of climate data.**
  - o Answer: Sentence was rephrased, to clarify, that is only applicable to our data.
  - o Changes in manuscript: 590
- **L 440: these perspectives should be cleraly detailed in the discussion. The reader needs to understand the research strategy and how the data will be used for the implementation of the perspective.**
  - o Answer: Discussion was rewritten with additional research strategy explained.
  - o Changes in manuscript: whole discussion part was rewritten.

**References**

Ewald, A., Hartmeyer, I., Keuschnig, M., Lang, A., and Otto, J.-C.: Fracture dynamics in an unstable, deglaciating headwall, Kitzsteinhorn, Austria, 1–25, https://doi.org/10.5194/tc-2019-42, 2019.

Gischig, V. S., Moore, J. R., Evans, K. F., Amann, F., and Loew, S.: Thermomechanical forcing of deep rock slope deformation: 2. The Randa rock slope instability, 116, https://doi.org/10.1029/2011JF002007, 2011.

Hasler, A., Gruber, S., and Haeberli, W.: Temperature variability and offset in steep alpine rock and ice faces, 5, 977–988, https://doi.org/10.5194/tc-5-977-2011, 2011.

Hasler, A., Gruber, S., and Beutel, J.: Kinematics of steep bedrock permafrost, 117, https://doi.org/10.1029/2011JF001981, 2012.

Krautblatter, M. and Moser, M.: A nonlinear model coupling rockfall and rainfall intensity based on a four year measurement in a high Alpine rock wall (Reintal, German Alps), 8, 2009.

Weber, S., Beutel, J., Da Forno, R., Geiger, A., Gruber, S., Gsell, T., Hasler, A., Keller, M., Lim, R., Limpach, P., Meyer, M., Talzi, I., Thiele, L., Tschudin, C., Vieli, A., Vonder Mühll, D., and Yücel, M.: A decade of detailed observations (2008–2018) in steep bedrock permafrost at the Matterhorn Hörnligrat (Zermatt, CH), Earth Syst. Sci. Data, 11, 1203–1237, https://doi.org/10.5194/essd-11-1203-2019, 2019.

---

## Author Response (AR2)

Dear editor

Thank you for all your comments and suggestions. Hopefully, all your questions and remarks will be answered in following text, or last version of manuscript. Answers to the comments follows.

**- Thank you very much for considering GI to publish your work. After receiving two reviews, for which I am very grateful, and receiving the revised manuscript the study can be considered for publication after minor revisions.**
**- The main issues are regarding the usage of the English language and the figures and tables. I suggest a thorough proof reading by a native speaker as there are typos and grammar issues throughout the manuscript making it sometimes not possible to understand what is meant (e.g. line 446-447). I tried to revise some, but I am also not a native speaker.**
Answer: Thank you for your suggestions and made revisions. We have rewritten the manuscript once more to improve English. Especially, changes were made in Discussion. Hopefully, these changes will make the manuscript more accessible for readers.
**- In its current form the manuscript still contains too many tables. Please, decrease number of tables and instead consider, which ones could and should be moved to the appendix or replaced by diagrams.**

Answer: Two tables were removed and two figures were merged. This revisions should help the manuscript, to be more comprehensible.

**- Please, check the captions of the tables and figures. They should enable that the figures and tables can be read by themselves, without looking into the manuscript. Furthermore, please, revise the tables and figures regarding their font size, as they are partly too small (e.g. tables 1 and 3 or figure 1, 2 and 6). And in general, please, thoroughly check the numbering of tables and figures, also in the text. For instance, figure 7 should be figure 9. And what table 7 are you referring to in line 325?**

Answer: Captions tables and figures were revisited and enhanced. Also tables font was enlarged in necessary cases. Hopefully now it is ok.
**- In table 2, I am wondering if the sampling rates of TLS and GB InSAR should be increased because measurements are also possible at minutes' intervals. Also, aerial photogrammetry can be measured with higher temporal resolution (hours-days if UAVs are considered). Furthermore, terrestrial photogrammetry enables minutes-days data capture if time-lapse systems are used.**

Answer: The table was updated according to your suggestions.
**- In figure 1, please remove the dots from the numbers in the figure.**
Answer: dots were removed and font was enlarged.
**- At the end of the introduction (line 55-77), please, reorder the paragraphs/sentences, e.g. based on geometric and environmental observations, because at the moment it seems to be a bit mixed up of mentioning of different monitoring options. Furthermore, please, rephrase the study site description because many sentences repeat (word by word) in each study site chapter. Also, please, explain the methods mentioned in line 235-237 in some more detail for readers not originating from geology. Finally, please, use consistent terminology and abbreviations, e.g. considering the geographic directions.**

Answer: Thanks for the suggestions. The paragraphs in introduction were reordered and we hope, that now the introduction is clearer. Also, description of monitoring sites was shortened and

rewritten. Same for the methods descriptions, which were enlarged. Terminology and abbreviations were revisited in final manuscript version.

Hopefully, revisions of manuscript will fulfil the expectations and standards of the Geoscientific Instumentation, Methods and Data Systems Journal.

In the name of all authors.

Ondřej Racek

---

## Editor Decision (ED2)

[revised manuscript text omitted]

---

## Author Response (AR3)

Dear editor

Thank you for your patience, comments and remarks. As you suggested, manuscript was corrected by native speaker, to improve readability and intelligibility. Hopefully these revisions and corrections will satisfy future readers.

In the name of all authors

Kind regards

Ondřej Racek